# Safety and Effectiveness of Carbon Dioxide Removal CO2RESET Device in Critically Ill Patients

**DOI:** 10.3390/membranes13070686

**Published:** 2023-07-24

**Authors:** Fabio Silvio Taccone, Simone Rinaldi, Filippo Annoni, Leda Nobile, Matteo Di Nardo, Jessica Maccieri, Anna Aliberti, Maximilan Valentin Malfertheiner, Andrea Marudi, Lars Mikael Broman, Mirko Belliato

**Affiliations:** 1Department of Intensive Care, Erasme Hospital, Lennik Road 808, 1070 Brussels, Belgium; filippo.annoni@erasme.ulb.ac.be (F.A.); leda.nobile@erasme.ulb.ac.be (L.N.); 2Struttura Complessa di Anestesia e Rianimazione, Ospedale Civile di Baggiovara, 41100 Modena, Italy; rinaldi.simone@aou.mo.it (S.R.); maccieri.jessica@aou.mo.it (J.M.); marudi.andrea@auo.mo.it (A.M.); 3Pediatric Intensive Care Unit, Bambino Gesù Children’s Hospital, IRCCS, 00165 Rome, Italy; matteo.dinardo@opbg.net; 4SC AR2 Anestesia e Terapia Intensiva Cardiotoracica, Foundation IRCCS Policlinico San Matteo, 27100 Pavia, Italy; a.aliberti@smatteo.pv.it (A.A.); m.belliato@gmail.com (M.B.); 5Internal Medicine II, University of Regensburg, 93053 Regensburg, Germany; maxmalfertheiner@gmail.com; 6ECMO Centre Karolinska, Karolinska University Hospital, 171 77 Stockholm, Sweden; lars.broman@regionstockholm.se; 7Department of Physiology and Pharmacology, Karolinska Institutet, 171 64 Stockholm, Sweden

**Keywords:** respiratory failure, CO_2_ removal, extracorporeal, high blood flow, ARDS, complications

## Abstract

Background: In this retrospective study, we report the effectiveness and safety of a dedicated extracorporeal carbon dioxide removal (ECCO_2_R) device in critically ill patients. Methods: Adult patients on mechanical ventilation due to acute respiratory distress syndrome (ARDS) or decompensated chronic obstructive pulmonary disease (dCOPD), who were treated with a dedicated ECCO_2_R device (CO2RESET, Eurosets, Medolla, Italy) in case of hypercapnic acidemia, were included. Repeated measurements of CO_2_ removal (VCO_2_) at baseline and 1, 12, and 24 h after the initiation of therapy were recorded. Results: Over a three-year period, 11 patients received ECCO_2_R (median age 60 [43–72] years) 3 (2–39) days after ICU admission; nine patients had ARDS and two had dCOPD. Median baseline pH and PaCO_2_ levels were 7.27 (7.12–7.33) and 65 (50–84) mmHg, respectively. With a median ECCO_2_R blood flow of 800 (500–800) mL/min and maximum gas flow of 6 (2–14) L/min, the VCO_2_ at 12 h after ECCO_2_R initiation was 157 (58–183) mL/min. Tidal volume, respiratory rate, and driving pressure were significantly reduced over time. Few side effects were reported. Conclusions: In this study, a dedicated ECCO_2_R device provided a high VCO_2_ with a favorable risk profile.

## 1. Introduction

Mechanical ventilation (MV) is a crucial intervention for patients with severe respiratory insufficiency who cannot maintain effective gas exchange through spontaneous breathing [1]. However, numerous detrimental consequences of MV have been identified, including lung over-inflation, oxygen toxicity, right ventricular failure, and ventilator-associated pneumonia (VAP), which have a profound impact on the management of ventilated patients [2,3]. In patients with acute respiratory failure, a ventilation strategy designed to minimize alveolar damage associated with MV, i.e., ventilator-induced lung injury (VILI), such as a low tidal volume approach, has been shown to effectively decrease mortality [4,5].

Recent studies have proposed a modern ventilatory approach (“mechanical load sparing”) based on the concept of protective ventilation, which includes small tidal volumes, low respiratory rates, and limited plateau pressures, even for patients with decompensated chronic obstructive pulmonary disease (COPD) in the peri- and post-operative setting [6,7]. However, this approach is challenging to implement due to the risk of hypercapnia, which is associated with pulmonary hypertension, decreased renal blood flow, and the release of endogenous catecholamines [8]. An adjuvant strategy, such as partial extracorporeal carbon dioxide removal (ECCO_2_R) using a dedicated device, appears to be an attractive solution to control hypercapnia in this setting [9].

ECCO_2_R involves an extracorporeal device operating on one or two venous access, a blood pump, a membrane lung, and a sweep gas system; the low blood flow (0.4–1.0 L/min) generated can provide CO_2_ removal without significant effect on blood oxygenation and target arterial CO_2_ (PaCO_2_) can be easily achieved by adjusting the sweep gas flow through the membrane lung [9,10]. However, for lower ranges of blood flow (i.e., 0.4 L/min), the membrane lung is often incorporated into continuous renal replacement therapy (CRRT) [9,11]. This implementation may be limited by the capability of CRRT devices to achieve such blood flows and by the smaller cannulas used for dialysis catheters, which could increase the risk of flow turbulence and hemolysis [12]. Moreover, lower CO_2_ extraction devices (i.e., low blood flow and low surface area) have been associated with a lower proportion of patients achieving adequate CO_2_ removal and a higher incidence of hemolysis and bleeding than higher CO_2_ extraction devices (i.e., higher blood flow and larger membrane surface area) [13]. Furthermore, full extracorporeal membrane oxygenation (ECMO) is frequently employed to perform CO_2_ removal at high ranges of blood flow (i.e., 1.0 L/min). However, this approach is associated with a similar rate of complications, economic burden, and organizational issues as ECMO use [14].

A new device has been recently made available on the market, which is entirely dedicated to ECCO_2_R with a reasonably high maximum blood flow (i.e., 0.8 L/min). This device produces a high CO_2_ removal rate of up to 150 mL/min in an animal model of induced hypercapnia [15]. However, human data on the effectiveness of such device are lacking. Therefore, the aim of this study was to assess the effectiveness and safety of a novel ECCO_2_R system in the removal of CO_2_ in patients with acute severe hypercapnia requiring MV.

## 2. Materials and Methods

### 2.1. Study Design

This was a retrospective multicentric study that included all adult (>18 years of age) patients treated with ECCO_2_R (CO2RESET—Eurosets, Medolla, Italy) in participating centers according to the decision of the treating physician for (a) acute respiratory distress syndrome (ARDS) or decompensated chronic obstructive pulmonary disease (dCOPD) and (b) pH < 7.25 and PaCO_2_ > 60 mmHg despite the optimal setting of MV. Separate ethics approval for this study was obtained from each participating center (i.e., Brussels, Modena and Pavia), and informed consent was waived because of the retrospective nature of the study. The study was conducted according to the requirements of the Declaration of Helsinki.

### 2.2. Intervention

CO2RESET is a CE-marked device that includes a new volumetric capnometer. It has a roller pump and a maximum blood flow of 800 mL/min. The membrane lung surface area is 1.81 m^2^ and the priming volume is 120 mL (Figure 1).

This device incorporates a mid-range infrared sensor placed in the mainstream, unlike standard devices that use side-stream sampling, at the membrane lung exhaust connector. Additionally, a flow sensor device is situated at the membrane lung sweep gas inlet. The concentration of CO_2_ is calculated based on the CO_2_ absorption characteristics of the radiation variation, similar to that of common capnometers. However, the combination of the capnometer and flowmeter allows for the continuous evaluation of the percentage of CO_2_ exhaust by the membrane lung, as well as the continuous calculation of CO_2_ removal (VCO_2_, mL/min). The VCO_2_ measurement has been validated using standard measurements [16]. Furthermore, a temperature control system placed inside the capnometer measuring cell maintains the temperature of the gas exhaust between 40 and 42 °C, preventing condensation and misleading measurements. Ultimately, dedicated software tools analyze the measurements and display the inlet gas flow, pre- and post-membrane lung pressures, and VCO_2_.

This medical equipment is intended for extracorporeal CO_2_ removal and is designed to be used in hypercapnic patients receiving mechanical ventilation. In all participating centers, CO2RESET was implanted percutaneously using either two heparin-coated cannulas or a single-vessel (i.e., double-lumen) cannulation, according to local practices and availability. The device was not used in case of severe hypoxemia, in case of contraindication to systemic anticoagulation, a body mass index (BMI) <15 or >40 kg/m^2^, or a moribund patient with do-not-resuscitate orders. CO2RESET priming consisted of a Plasmalyte solution (Baxter Healthcare Corporation, Deerfield, MA, USA). The monitoring of heart and lung function, hemodynamic targets, the initiation of anticoagulants and anticoagulation targets, and thresholds for red blood cell transfusion (RBCT) were left at the discretion of the treating team. Device weaning was decided according to local protocols.

### 2.3. Data Collection

We gathered demographic data, pre-existing chronic disease history, and admission diagnosis of all patients. The severity of illness was evaluated using the Acute Physiology and Chronic Health Evaluation (APACHE) II score [17] on ICU admission and the Sequential Organ Failure Assessment (SOFA) [18] score on the day ECCO_2_R was initiated. We documented the requirement for MV, vasoactive agents, anticoagulants, and blood transfusions, as well as the duration of ICU stay and overall ICU and hospital mortalities. ECCO_2_R data were collected, including indications, configuration, weaning, circuit function, and biological data, on the day of initiation and 24 h after implementation. Arterial blood gas analyses, MV settings, and the use of sedatives and/or neuromuscular blocking agents (NMBAs) at the beginning of ECCO_2_R, and at 1, 12 (±1), and 24 (±2) h thereafter were also recorded, when feasible. These time points were selected because they were available in all participating centers according to the local monitoring protocols.

### 2.4. Patients’ Selection and Study Outcomes

The primary outcome was to assess the maximum VCO_2_ value within the first 12 h of therapy. Secondary outcomes included (a) hospital mortality; (b) weaning from MV; (c) the occurrence of any clinical adverse events, such as moderate or severe bleeding, red blood cell transfusion (RBCT), any cardiac arrhythmias, hypotension, the need for vasopressors, and the occurrence of cardiac arrest; (d) the occurrence of adverse events associated with the extracorporeal device, including kinking of tubing, clotting, accidental decannulation, membrane lung failure, pump failure, and circuit damage.

### 2.5. Statistical Analysis

As this was a retrospective study, no sample size calculation was initially made. However, considering that the expected maximum VCO_2_ from CO2RESET of 100 (±15) mL/min would be superior to the VCO_2_ values achieved with other low blood flow devices, i.e., around 45 (±10) mL/min [12,13,19,20], we calculated a posteriori that a sample size of at least 9 patients was, therefore, required to achieve 90% power in detecting a superiority of CO2RESET for the primary endpoint. Data are presented as the median (ranges) or count (%). Data from repeated measures were analyzed using the Kruskal–Wallis test, followed by the Bonferroni post hoc analysis to evaluate differences between each time point. For biological measurements, a Wilcoxon matched-pairs signed rank test was used. All the statistical analyses for this study were performed in R version 3.6.0 (R Foundation for Statistical Computing, Vienna, Austria). The *p*-values were two-tailed and values < 0.05 were considered statistically significant.

## 3. Results

### 3.1. Study Population

Between January 2020 and December 2022, a total of 11 patients received ECCO_2_R treatment at three participating medical centers (Brussels, n = 8; Modena, n = 2; Pavia, n = 1). The primary baseline characteristics are presented in Table 1. The median age of the patients was 60 (43–72) years. The median APACHE II score upon admission and SOFA score on the day of ECCO_2_R initiation were 19 (16–24) and 9 (7–10), respectively. Among the patients, nine had ARDS (six with COVID-19 and three with sepsis), while two had dCOPD (one with bacterial infection and the other with viral pulmonary infection). The median times from ICU admission to MV and from ICU admission to ECCO_2_R initiation were 1 (1–3) days and 3 (2–39) days, respectively. None of the patients required CRRT during the ECCO_2_R therapy.

### 3.2. ECCO_2_R Implementation

The median baseline pH and PaCO_2_ levels were 7.27 (7.12–7.33) and 65 (50–84) mmHg, respectively. The median baseline ventilatory settings are presented in Table 2; the median driving pressure was 21 (14–24) cmH_2_O. Two patients failed prone positioning to improve hypercapnia and ventilatory mechanics, and nine patients were on norepinephrine (median dose of 8 [3–12] mcg/min). At baseline, all patients were receiving continuous infusion of sedatives and opioids, and 10 (91%) patients were administered NMBAs.

Regarding configuration, all patients received dual-site cannulation, with the drainage cannula inserted via the femoral (n = 7) or left subclavian (n = 4) vein, with a median cannula size of 18 (16–21) Fr. A return cannula was inserted via the right jugular vein in all patients, with a median cannula size of 15 (15–18) Fr. The median duration of ECCO_2_R was 6 days (3–17) days. Anticoagulation was initiated immediately after ECCO_2_R implantation using continuous intravenous infusion of unfractionated heparin, targeting an activated clotting time of 160–200 s, as measured by the available devices in each center.

### 3.3. Primary Outcome

With a median ECCO_2_R blood flow of 800 (500–800) mL/min and a maximum sweep gas flow of 6 (2–14) L/min, VCO_2_ at 12 h after ECCO_2_R initiation was 157 (58–183) mL/min. Compared to the values reported for low-flow ECCO_2_R devices (i.e., 45 ± 10 mL/min), the observed VCO_2_ values were significantly higher (*p* = 0.009). The maximum VCO_2_ over the first 24 h was 161 (72–188) mL/min; VCO_2_ values over time are presented in Figure 2.

Tidal volume, respiratory rate, and driving pressure significantly decreased over time when compared to the baseline values (Table 2 and Figure 3). At 24 h, pH was 7.41 (7.35–7.50) and PaCO_2_ was 45 (39–49) mmHg. The biological changes at 24 h after ECCO_2_R initiation are reported in Table 2.

### 3.4. Secondary Outcomes

The ECCO_2_R device was successfully weaned in seven patients (64%). Hospital mortality was observed in seven patients (64%, including three of those in whom ECCO_2_R was weaned), mainly because of multiple organ failure (n = 3), refractory hypoxemia (n = 3; all patients were not eligible for ECMO), and septic shock (n = 1).

In three patients, a second circuit was required either due to membrane lung clotting (n = 2), occurring after 5 and 6 days, or circuit expiration (n = 1), happening after 7 days. One instance of device malfunction was observed in the first treated patient, attributed to a software bug in the heparin syringe calibration, which was eventually resolved without injury to the patient. No pump failure, circuit damage, catheter kinking, or dislodgment was reported. Three patients developed hematomas at the drainage cannulation site, and two of them required red blood cell transfusions (1 and 4 packs) due to bleeding. No cases of air embolism or catheter infection were reported. None of the patients experienced cardiac arrhythmia or cardiac arrest during ECCO_2_R treatment. No additional patients required vasopressors in the 48 h following ECCO_2_R initiation.

## 4. Discussion

The results of this study suggest that ECCO_2_R, using a dedicated device, is an effective and safe method for managing hypercapnic acidemia in critically ill patients with ARDS or dCOPD. Also, higher blood flow and VCO_2_ are important advantages over ECCO_2_R devices used with CRRT. The significant reduction in ventilatory mechanical load observed over time highlights the potential of ECCO_2_R to reduce ventilator-induced lung injury. Moreover, the low incidence of adverse events associated with CO2RESET in this study is encouraging.

In individuals experiencing acute respiratory failure, ECCO_2_R is often employed as an adjunct to MV to provide additional support for gas exchange [19]. ECCO_2_R therapy can be used in critically ill patients with ARDS to facilitate ultra-protective lung ventilation (UPLV) and decrease VILI by reducing tidal volume, driving pressures, and respiratory rate, while also controlling respiratory acidosis [19,20,21]. Additionally, ECCO_2_R therapy may be implemented in individuals with dCOPD exhibiting severe respiratory acidosis and hypercapnic respiratory failure to avoid intubation in those at risk of non-invasive ventilation failure or to expedite weaning and promote early extubation in individuals requiring MV [22]. Combes et al. demonstrated that the UPLV strategy was feasible in 78% and 82% of patients with moderate ARDS at 8 and 24 h after implementation [21]. However, a secondary analysis of this database showed that UPLV was more frequently achieved in patients with devices having high VCO_2_ and blood flow compared to lower blood flow, which significantly questioned the role of ECCO_2_R implemented in CRRT devices in such patients [13]. A recent randomized trial including 412 adult patients receiving MV for acute hypoxemic respiratory failure and randomized to early ECCO_2_R for UPLV was stopped early due to futility, as ECCO_2_R to facilitate lower tidal volume ventilation did not significantly reduce 90-day mortality compared to conventional ventilation strategies [23]. Although our results do not add evidence on the effectiveness of ECCO_2_R on patient-relevant outcomes or on the optimal timing and eligibility criteria to initiate such therapy, we demonstrated that a dedicated ECCO_2_R device can provide a very high VCO_2_, which will result in a rapid improvement in gas exchange and ventilatory settings. The significant reduction in tidal volume, driving pressure, and respiratory rate resulted in more protective lung ventilation. Although we did not specifically calculate the mechanical power related to mechanical ventilation nor assess the long-term effects of this ECCO_2_R device, such a solution could be a suitable and balanced approach to avoid ineffective CO_2_ removal from low blood flow devices and, perhaps, to avoid the need of ECMO support later on, with a reduced risk of potential side effects.

The membrane of the lung plays a crucial role in extracorporeal CO_2_ removal. In particular, the surface area of the membrane is an important factor in the ECCO_2_R device efficacy, as it directly affects the efficiency of gas exchange [24]. A larger surface area allows for greater contact between the blood and gas phase, facilitating the diffusion of CO_2_ and oxygen across the membrane. Polymeric hollow fiber membranes can provide a much larger surface area than traditional gas absorption towers or liquid extraction columns; the structure of these membranes is designed to be thin and porous, enabling rapid gas exchange while maintaining adequate mechanical strength [25]. As different types of membranes are available for ECCO_2_R devices, they vary in their characteristics and performance. Some commonly used membranes include polymethylpentene (PMP), polypropylene hollow fibers, and silicone membranes. These membranes differ in their gas permeability, biocompatibility, durability, and resistance to clot formation [26]. The membrane used in the CO2RESET device is more efficient than other low-flow devices in eliminating CO_2_ while maintaining similar biocompatibility. Moreover, higher blood flow can increase CO_2_ exchange and minimize the risk of clot formation [27]. Additionally, the durability of the membrane is essential to ensure the longevity of the ECCO_2_R device. However, it is important to acknowledge that some low-flow devices are available to provide much higher CO_2_ removal rates (i.e., 75 mL/min), although some data are only related to in vitro models [25]. Furthermore, it is worth noting that other devices have demonstrated VCO_2_ values of approximately 140 mL/min when utilizing a blood flow rate of 750 mL/min [28]. However, it is important to acknowledge that these findings originated from an animal study and have not yet been replicated in human subjects. Therefore, clinicians must carefully consider the characteristics and efficacy of the ECCO_2_R device when selecting it for a patient, taking into account the anticipated impact on gas exchange in the treated individual.

Another important finding is the relatively low occurrence of side effects with the CO2RESET device [22,29]. In a previous study, ECCO_2_R-related episodes of hemolysis and bleeding were higher with lower extraction (n = 33) devices than with higher extraction (n = 62) devices, and adverse events were reported in 39% of patients, including two severe adverse events directly attributed to ECCO_2_R [13]. Additionally, ECCO_2_R has been associated with a significantly lower number of ventilator-free days compared to standard care, and serious adverse events were reported in 31% of ECCO_2_R patients [23]. The reasons for the differences with the existing literature, whether related to the characteristics of the device, the use of higher blood flow, better local anticoagulation practices, patient selection, and/or the use of double cannulation, remain unclear. However, given the inconclusive evidence of its effectiveness and the relatively high number of adverse events reported in previous studies, CO2RESET may represent an appropriate choice to achieve a satisfactory risk/benefit balance with this approach. One patient encountered a severe bleeding incident at the cannulation site, necessitating the administration of four units of red blood cell transfusions. However, it is important to note that complications associated with cannulation are not specific to the CO2RESET device, as they can potentially occur with any extracorporeal therapy.

This study has several limitations that should be acknowledged. First, it is important to note that the study had a small sample size, and larger studies are needed to confirm these findings. The results may also be dependent on local experience, may limit their generalizability, and require external validation. Second, other side effects, such as hemolysis and thrombocytopenia, were not systematically collected. While the definition of hemolysis in this setting remains controversial [30] and free hemoglobin was not homogeneously measured in the participating centers, thrombocytopenia may be due to other mechanisms, such as sepsis, drug reactions, or bleeding, and may provide difficult-to-interpret findings in the absence of a control group. Third, we did not include a parallel control group using a low blood flow device. However, the measured VCO_2_ with CO2RESET was significantly higher than the reported rates with other devices, and the configuration of those devices (i.e., cannula size and blood flow) limited their ability to obtain higher CO_2_ removal rates. Finally, only data at 24 h were collected. Future studies on the CO2RESET device should focus on several days of therapy to understand whether the CO_2_ removal rate remains stable over the entire operation period.

## 5. Conclusions

In this study, the use of the CO2RESET device showed to be effective for CO_2_ elimination in selected ARDS and dCOPD patients, with few side effects. Future studies are required to describe its effectiveness in fewer selected patients, as well as other potential side effects and pitfalls.

## Figures and Tables

**Figure 1 membranes-13-00686-f001:**
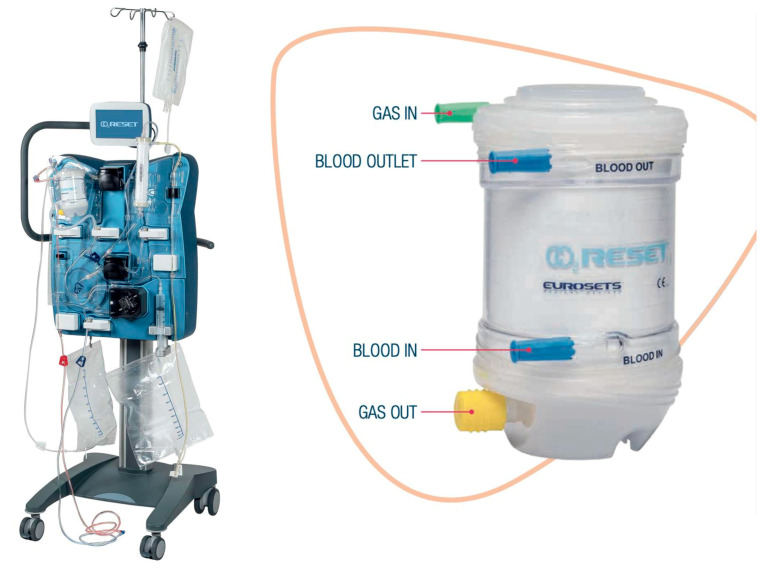
Representation of the CO2RESET device (**left**) and the membrane lung (**right**).

**Figure 2 membranes-13-00686-f002:**
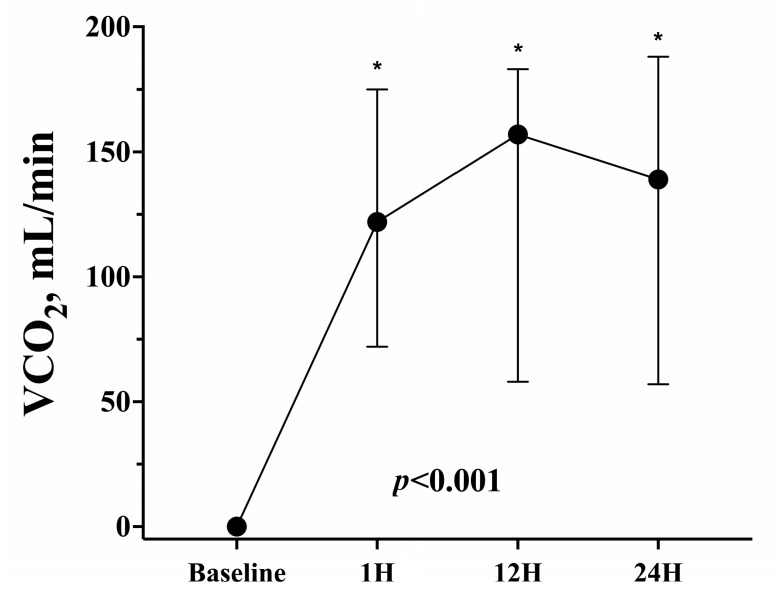
CO_2_ removal (VCO_2_) over time. * indicates significant differences with baseline values.

**Figure 3 membranes-13-00686-f003:**
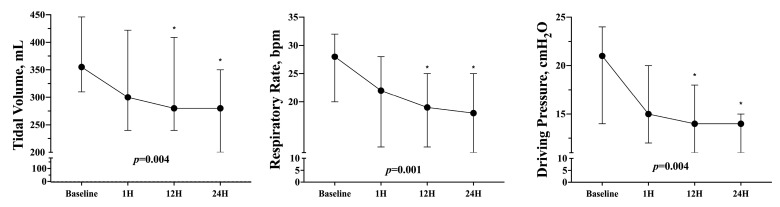
Tidal volume, respiratory rate, and driving pressure change over time. * indicates significant differences with baseline values.

**Table 1 membranes-13-00686-t001:** Characteristics of the study population. Data are reported as count (%) or median (ranges).

Characteristic	N = 11
Age, years	60 (43–72)
Male sex—n (%)	9 (82)
Weight, Kgs	88 (62–160)
APACHE II score on admission	19 (16–24)
** *Medical History* **	
Chronic Heart Disease—n (%)	0
Coronary Artery Disease—n (%)	0
COPD, n (%)	2 (18)
Diabetes—n (%)	1 (9)
Immunosuppression—n (%)	2 (18)
** *ECCO_2_R characteristics* **	
Time from admission to MV, days	1 (1–3)
Time from admission to ECCO_2_R, days	3 (2–39)
SOFA Score on the day of ECCO_2_R	9 (7–10)
Duration of therapy, days	6 (3–17)
** *Adverse Events* **	
Circuit Change, n (%)	3 (27)
Device Malfunctioning, n (%)	1 (9)
Catheter Kinking, n (%)	0
Catheter Dislodgement, n (%)	0
Cannulation Site Hematoma, n (%)	3 (27)
Bleeding requiring RBCT, n (%)	2 (18)
Air Embolism, n (%)	0
Cannula infection, n (%)	0
Arterial Hypotension, n (%)	0
New onset of vasopressors, n (%)	0
** *Outcomes* **	
ICU length of stay, days	19 (8–71)
Tracheostomy, n (%)	3 (27)
Hospital mortality, n (%)	7 (64)

COPD = chronic obstructive pulmonary disease; ECCO_2_R = extracorporeal CO_2_ removal; SOFA = sequential organ failure assessment; ICU = intensive care unit.

**Table 2 membranes-13-00686-t002:** Time course of main study variables, including biological testing. Data are presented as median (ranges) or count (%).

Variable	Baseline	1-h	12-h	24-h	*p* Value
**pH**	7.27 (7.12–7.33)	7.40 (7.30–7.50)	7.40 (7.35–7.48)	7.41 (7.35–7.50)	<0.001
**PaCO_2_, mmHg**	65 (50–84)	50 (31–63)	47 (32–63)	45 (39–49)	<0.001
**PaO_2_, mmHg**	94 (70–129)	90 (58–207)	90 (67–131)	91 (72–118)	0.83
**ECCO_2_R BF, mL/min**	0 (0–0)	800 (500–800)	800 (500–800)	800 (550–800)	<0.001
**ECCO_2_R GF, L/min**	0 (0–0)	4 (2–14)	6 (1.5–14)	6 (1–14)	<0.001
**VCO_2_, mL/min**	0 (0–0)	122 (72–175)	157 (58–183)	139 (57–188)	<0.001
**FiO_2_, %**	60 (40–100)	60 (40–90)	50 (40–95)	50 (40–100)	0.85
**PEEP, cmH_2_O**	8 (4–12)	8 (4–14)	8 (4–12)	8 (4–14)	0.93
**TV, mL**	355 (310–446)	300 (240–422)	280 (240–409)	280 (200–350)	0.004
**TV/PBW, mL/kg**	4.9 (4.4–6.0)	4.1 (3.7–5.3)	3.9 (3.3–5.4)	3.8 (2.5–4.6)	0.001
**RR**	28 (20–32)	22 (12–28)	19 (12–25)	18 (11–25)	0.001
**Driving Pressure, cmH_2_O**	21 (14–24)	15 (12–20)	14 (11–18)	14 (11–15)	0.004
**NE dose, mcg/min**	8 (3–12), n = 9	10 (3–19), n = 9	8 (2–22), n = 9	8 (5–18), n = 7	0.67
**Sedatives, n (%)**	11 (100)	11 (100)	11 (100)	11 (100)	0.99
**Opioids, n (%)**	11 (100)	11 (100)	11 (100)	11 (100)	0.99
**NMBA, n (%)**	10 (91)	10 (91)	10 (91)	10 (91)	0.99
**Fibrinogen, mg/dL**	460 (178–815)	-	-	457 (210–777)	0.15
**D-dimers, ng/mL**	2100 (758–22,327)	-	-	1650 (1077–15,000)	0.69
**Hemoglobin, g/dL**	10.1 (8.9–13.9)	-	-	9.0 ((8.1–12.2)	0.10
**Platelets, /mm^3^*10^3^**	210 (118–511)	-	-	188 (112–410)	0.18
**LDH, IU/L**	350 (155–515)	-	-	311 (189–550)	0.57

ECCO_2_R = extracorporeal CO_2_ removal; BF = blood flow; GF = gas flow; VCO_2_ = CO_2_ removal; FiO_2_ = inspired oxygen fraction; PEEP = positive end-expiratory pressure; TV = tidal volume; RR = respiratory rate; NMBA = neuromuscular blocking agents; LDH = lactate dehydrogenase; PBW = predicted body weight.

## Data Availability

Data are available upon request from the corresponding author.

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
