# Peer review of "Safety and Effectiveness of Carbon Dioxide Removal CO2RESET Device in Critically Ill Patients"

_membranes, 2023, doi:10.3390/membranes13070686_

Round 1
Reviewer 1 Report
I would like to thank the editor and the authors for the opportunity to read this interesting manuscript. Taccone et al. describe their experiences with a new ECCO2R device in a small sample size of patients and provide physiological data including adverse events in 11 patients. The manuscript is of interest in the scope of the journal.
I´d like to make a couple of remarks in the hope to increase the overall understandability of the manuscript.
1. I have difficulties to understand, where the data is actually coming from. It is difficult to understand the power analysis provided by the authors in the context of the retrospective character of the data mentioned in line 80. The authors should mention the fact that this is a retrospective analysis of prospectively acquired data in the abstract.
2. I am interested in the patient selection for the ECCO2R device.
It uses two sides of venous access like a proper ECMO device….so no real advantage there.
The patients were also rightfully put on the device relatively early in the treatment on ICU. In my mind, at this timepoint it is difficult to estimate the trajectory the patient is taking during the ICU…i.e., from hypercapnic to a combination of hypoxic/hypercapnic respiratory failure. So perhaps the authors could clarify their approach to patient selection for ECCO2R compared to ECMO in the Materials and Methods and the Discussion. This is of special interest as the authors studied comparable few patients in the timespan of 3 years.
3. How much patients were on CRRT necessitating a second extracorporeal circuit compared to ECMO where you can simply plug in the CRRT in the existing circuit? Or is
Minor remarks:
1. Introduction: line 71: A recent device has been recently made available…please correct
2. Material and Methods: line 88: please correct the typo
3. Material and Methods: line 110-113: the sentence appears to be misleading. The device was not used, if the patients was hypoxemic?
4. Material and Methods: line 110-113: why was the device not used in patients with a BMI > 40? According to a recent analysis of the ELSO data base, putting obese patients on ECMO seems to be international practice in many centers (PMID: 36416896)
5. Table 2 seems to need a revision of the headlines
6. Table 2: the presentation of TV; RR, and Pdriv in table 2 and figure 3 seems to be redundant.
Author Response
- Taccone et al. are presenting a retrospective study in mechanically ventilated ARDS and dCOPD patients investigating the effectiveness and safety of a medium-flow ECCO2R device with a median blood flow rate of 800 ml/min. Although, the study has limitations in particular with respect to patient numbers and the absence of a control, the findings of the manuscript are of interest for the reader. The study could demonstrate that high CO2 removal rates can be achieved with ECCO2R devices under the applied settings, which resulted in a correction of hypercapnia and respiratory acidosis and enabled a reduction in tidal volume, respiratory rate and driving pressure. The content is technically sound, reflecting best practices in this area of study. The endpoints and methods used are appropriate. The manuscript is well-written and easy to read, with a good balance of text, tables, and illustrating figures. I have no concerns and recommend acceptance after the minor points listed below have been addressed.
Authors’ response: We thank the reviewer for the positive comments.
- Page 2, line 64 - 66 “lower ECCO2R blood flows have been associated with lower proportion of patients achieving adequate CO2 removal….”: SUPERNOVA study compared lower CO2 extraction devices (low blood flow and low surface area) with higher CO2 extraction devices (higher blood flow and larger membrane surface area). Both aspects blood flow and surface area contributed to the difference in outcome. Please clarify.
Authors’ response: We have added this point to the text, as suggested.
- Page 4, Line 143: “… superior to VCO2 values achieved with other low blood flow devices, i.e. around 45 (+- 10) ml/min…”: Hospach et al. (Hospach, I., et al. (2020). "In vitro characterization of PrismaLung+: a novel ECCO2R device." Intensive Care Medicine Experimental 8(1): 14) presented a low flow ECCO2R device achieving “CO2 removal rates …[of] 73 ± 4.0 mL/min…. at [blood flow] QB 300 mL/min and pinCO2 45mmHg.” Please take into consideration that there are low flow devices available with much larger CO2 removal rates than 45 ml/ml, which you used for your sample size calculation.
Authors’ response: We understand the point of the reviewer and have added this as a comment into the Discussion (the same has been raised by the reviewer 2).
- PAGE 6; Table 2: Table header row is missing in the version I’ve received. Please add. PEEP: please add unit (PEEP, cm H2O). TV: for better comparability with data from other studies please normalize tidal volume to predicted body weight.
Authors’ response: This has been modified, accordingly.
- The median duration of ECCO2R was 6 days (3-17). Please comment why you’ve presented only data for the first 24 hours. Data on CO2 removal after several days would show whether the removal rate is stable over the entire operation period of the device.
Authors’ response: This has been added as a Limitation. We only collected data in the early phase of CO2 removal, to understand the effectiveness of the device but did not harmonize data collection for the following days.
Reviewer 2 Report
Taccone et al. are presenting a retrospective study in mechanically ventilated ARDS and dCOPD patients investigating the effectiveness and safety of a medium-flow ECCO2R device with a median blood flow rate of 800 ml/min. Although, the study has limitations in particular with respect to patient numbers and the absence of a control, the findings of the manuscript are of interest for the reader. The study could demonstrate that high CO2 removal rates can be achieved with ECCO2R devices under the applied settings, which resulted in a correction of hypercapnia and respiratory acidosis and enabled a reduction in tidal volume, respiratory rate and driving pressure. The content is technically sound, reflecting best practices in this area of study. The endpoints and methods used are appropriate. The manuscript is well-written and easy to read, with a good balance of text, tables, and illustrating figures. I have no concerns and recommend acceptance after the minor points listed below have been addressed.
Page 2, line 64 - 66 “lower ECCO2R blood flows have been associated with lower proportion of patients achieving adequate CO2 removal….”: SUPERNOVA study compared lower CO2 extraction devices (low blood flow and low surface area) with higher CO2 extraction devices (higher blood flow and larger membrane surface area). Both aspects blood flow and surface area contributed to the difference in outcome. Please clarify.
Page 4, Line 143: “… superior to VCO2 values achieved with other low blood flow devices, i.e. around 45 (+- 10) ml/min…”: Hospach et al. (Hospach, I., et al. (2020). "In vitro characterization of PrismaLung+: a novel ECCO2R device." Intensive Care Medicine Experimental 8(1): 14) presented a low flow ECCO2R device achieving “CO2 removal rates …[of] 73 ± 4.0 mL/min…. at [blood flow] QB 300 mL/min and pinCO2 45mmHg.” Please take into consideration that there are low flow devices available with much larger CO2 removal rates than 45 ml/ml, which you used for your sample size calculation.
PAGE 6; Table 2:
- - Table header row is missing in the version I’ve received. Please add
- - PEEP: please add unit (PEEP, cm H2O)
- - TV: for better comparability with data from other studies please normalize tidal volume to predicted body weight
The median duration of ECCO2R was 6 days (3-17). Please comment why you’ve presented only data for the first 24 hours. Data on CO2 removal after several days would show whether the removal rate is stable over the entire operation period of the device.
Author Response
- Thank you for the invitation to review this manuscript. The authors describe a series of 11 patients who received ECCO2R with a “medium” flow dedicated device, the CO2reset. The study is retrospective in nature and describe the performance of the device in achieving more protective ventilation targets in patients with ARDS or dCOPD. As the paper stands, I have some concerns for the authors to consider.
Authors’ response: We thank the reviewer for the time dedicated to revise our manuscript.
- I find the ethics perplexing. The author obtained waiver of consent because of the retrospective nature of the study, yet on page 4 (line 117-119) they state that all centers agreed to assess gas analysis at 1,12 and 24 hours, and collect CO2RESET data in a patient management system. This is not consistent with a retrospective study. The author need to clarify this aspect of the methods. Perhaps the blood gases were part of an agreed protocol for CO2RESET use or something? It cannot be that the blood gases were collected for this retrospective study, that breaches conventional study ethics.
Authors’ response: We understand the concerns of the reviewer – indeed, the study was entirely retrospective and we have explained that the time-points were selected as available in all centers according to local protocols. Being among the first centers to have the CO2RESET available in clinical practice, this approach has been decided to protocolize clinical management and not for the present study. We hope to have clarified this issue.
- I don’t understand methods and the use of double negative page 3, line 110, “the device was not use when ECMO was not indicated”, what does this mean? Only patients who would be eligible for ECMO received the device, or patient who were eligible for ECMO received the device, which would be odd, since indications for ECMO are quite different to those for ECCO2R. I think the authors are trying to describe contraindications here perhaps, but it is unclear and needs rewriting.
Authors’ response: We agree this is misleading. This has been removed, accordingly.
- I think the article will benefit from the authors highlighting more clearly what targets were met with the new device to give the manuscript and study more meaning. VCO2 alone is not enough (See number 4 below).
Authors’ response: Our primary aim was based on the efficacy of the device (i.e. VCO2). We have also highlighted some additional findings, such as the possibility to reduce tidal volume and respiratory rate, thus providing a more protective lung ventilation. This has been mentioned in the Discussion.
- When comparing CO2 removal of other devices, (line 189, results) one should reference where the 45 +/- 10mL/min came from and what the blood flow was. Godet et al, Crit Care 2014 report 40-60 ml/min for flows around 400 (PRismalung) and Karagiannidis et al, Crit Care 2014 reported 60-80 ml/min for flows 200-400. Interestingly, Karagiannidis reported CO2 removal of 138.3 ± 22.8 at blood flows of 750mL/min. So these results are in line what would be expected, reducing the novelty of the findings, hence addressing number 3 above will improve the impact. In addition, this comment on the results, i.e. comparing to other devices, should be in the discussion.
Authors’ response: We understand the point of the reviewer and have added this as a comment into the Discussion (the same has been raised by the reviewer 1). Importantly, the suggested study was conducted in animals and not replicated in humans.
- Some clumsy sentence construction, E.g. “recent device has recently..” (pg 2, line 71-73) and over use of phrase “As such” in discussion.
Authors’ response: the text has been revised, accordingly.
- Clumsy repeat of “the device” lines 90 and 94.
Authors’ response: The text has been changed, accordingly.
- Multiple typos/errors throughout Eg, reasonable instead of reasonably, repeat of could (line 244-245), provide instead of provided (line 263). Most can be corrected with a thorough proof-reading by the authors.
Authors’ response: the text has been revised, accordingly.
- Table 2 columns are not labeled, presume they are baseline, 1hr, 12hr and 24hr.
Authors’ response: This has been modified, accordingly.
Reviewer 3 Report
Thank you for the invitation to review this manuscript. The authors describe a series of 11 patients who received ECCO2R with a “medium” flow dedicated device, the CO2reset. The study is retrospective in nature and describe the performance of the device in achieving more protective ventilation targets in patients with ARDS or dCOPD. As the paper stands, I have some concerns for the authors to consider.
Major
(1) Methods. I find the ethics perplexing. The author obtained waiver of consent because of the retrospective nature of the study, yet on page 4 (line 117-119) they state that all centres agreed to assess gas analysis at 1,12 and 24 hours, and collect CO2RESET data in a patient management system. This is not consistent with a retrospective study. The author need to clarify this aspect of the methods. Perhaps the blood gases were part of an agreed protocol for CO2RESET use or something? It cannot be that the blood gases were collected for this retrospective study, that breaches conventional study ethics.
(2) I don’t understand methods and the use of double negative page 3, line 110, “the device was not use when ECMO was not indicated”, what does this mean? Only patients who would be eligible for ECMO received the device, or patient who were eligible for ECMO received the device, which would be odd, since indications for ECMO are quite different to those for ECCO2R. I think the authors are trying to describe contraindications here perhaps, but it is unclear and needs rewriting.
(3) I think the article will benefit from the authors highlighting more clearly what targets were met with the new device to give the manuscript and study more meaning. VCO2 alone is not enough (See number 4 below).
(4) When comparing CO2 removal of other devices, (line 189, results) one should reference where the 45 +/- 10mL/min came from and what the blood flow was. Godet et al, Crit Care 2014 report 40-60 ml/min for flows around 400 (PRismalung) and Karagiannidis et al, Crit Care 2014 reported 60-80 ml/min for flows 200-400. Interestingly, Karagiannidis reported CO2 removal of 138.3 ± 22.8 at blood flows of 750mL/min. So these results are in line what would be expected, reducing the novelty of the findings, hence addressing number 3 above will improve the impact. In addition, this comment on the results, i.e. comparing to other devices, should be in the discussion.
Minor
(1) Some clumsy sentence construction, E.g. “recent device has recently..” (pg 2, line 71-73) and over use of phrase “As such” in discussion.
(2) Clumsy repeat of “the device” lines 90 and 94.
(3) Multiple typos/errors throughout Eg, reasonable instead of reasonably, repeat of could (line 244-245), provide instead of provided (line 263). Most can be corrected with a thorough proof-reading by the authors.
(4) Table 2 columns are not labeled, presume they are baseline, 1hr, 12hr and 24hr.
See comments to author
Author Response
- I would like to thank the editor and the authors for the opportunity to read this interesting manuscript. Taccone et al. describe their experiences with a new ECCO2R device in a small sample size of patients and provide physiological data including adverse events in 11 The manuscript is of interest in the scope of the journal. I ́d like to make a couple of remarks in the hope to increase the overall understandability of the manuscript.
Authors’ response: We thank the reviewer for the kind comment.
- I have difficulties to understand, where the data is actually coming from. It is difficult to understand the power analysis provided by the authors in the context of the retrospective character of the data mentioned in line 80. The authors should mention the fact that this is a retrospective analysis of prospectively acquired data in the abstract.
Authors’ response: We thank the reviewer for this comment. Of course, this is a retrospective study. As we considered to compare VCO2 values from CO2RESET to values reported in other low-flow devices, we tried to make a post hoc analysis of the power of the study. This has been properly presented, accordingly. Abstract has been modified.
- I am interested in the patient selection for the ECCO2R device. It uses two sides of venous access like a proper ECMO device....so no real advantage there. The patients were also rightfully put on the device relatively early in the treatment on ICU. In my mind, at this timepoint it is difficult to estimate the trajectory the patient is taking during the ICU...i.e., from hypercapnic to a combination of hypoxic/hypercapnic respiratory failure. So perhaps the authors could clarify their approach to patient selection for ECCO2R compared to ECMO in the Materials and Methods and the Discussion. This is of special interest as the authors studied comparable few patients in the timespan of 3 years.
Authors’ response: The decision to initiate ECCO2R was made by the treating team. This is the main limitation of this small and selected cohort (as reported in the Discussion). We have clearly stated that this is a study focusing on VCO2, not on clinical outcomes. The double site cannulation was chosen because of the high costs of a double-lumen cannula.
- How much patients were on CRRT necessitating a second extracorporeal circuit compared to ECMO where you can simply plug in the CRRT in the existing circuit?
Authors’ response: No patient was also on CRRT. This has been added.
- Introduction: line 71: A recent device has been recently made available...please correct
Authors’ response: This has been corrected.
- Material and Methods: line 88: please correct the typo
Authors’ response: This has been corrected, accordingly.
- Material and Methods: line 110-113: the sentence appears to be misleading.
Authors’ response: This has been corrected, accordingly.
- The device was not used, if the patients was hypoxemic?
Authors’ response: This has been corrected, accordingly.
- Material and Methods: line 110-113: why was the device not used in patients with a BMI > 40? According to a recent analysis of the ELSO data base, putting obese patients on ECMO seems to be international practice in many centers (PMID: 36416896)
Authors’ response: This decision was based on the potential limitation due to an inadequate blood flow of the device when compared to the total body weight of obese patients.
- Table 2 seems to need a revision of the headlines
Authors’ response: This has been added, accordingly.
- Table 2: the presentation of TV; RR, and Pdriv in table 2 and figure 3 seems to be redundant.
Authors’ response: as these data are very important and numbers are not available in the Figure, we prefer to keep this information in the Table.
Round 2
Reviewer 3 Report
The authors have made an effort to address the reviewers comments. The text at line 118-120, requires "case" to be replaced by "cases", and the new text at line 138 needs rewriting for clarity as well.
There are some writing hiccups still present that might be addressed at copy editing stage, or by asking the authors to carefully proof read their manuscript.